# Synchronization Theory-Based Analysis of Coupled Vibrations of Dual-Tube Coriolis Mass Flowmeters

**DOI:** 10.3390/s20216340

**Published:** 2020-11-06

**Authors:** Zhong-Xiang Li, Chun Hu, De-Zhi Zheng, Shang-Chun Fan

**Affiliations:** 1School of Instrumentation and Optoelectronic Engineering, Beihang University, Beijing 100191, China; lingfly@buaa.edu.cn (Z.-X.L.); zhengdezhi@buaa.edu.cn (D.-Z.Z.); fsc@buaa.edu.cn (S.-C.F.); 2School of Electronic and Information Engineering, Beihang University, Beijing 100191, China; 3Research Institute for Frontier Science, Beihang University, Beijing 100191, China

**Keywords:** Coriolis mass flowmeter, coupled vibrations, nonlinear dynamics, frequency-doubling signal, modal analysis

## Abstract

Certain nonlinear influences are found in dual-tube Coriolis mass flowmeters (CMFs). According to experimentation, a nonlinearity dominated by frequency-doubling signals can be observed in the measuring signal. In general, such nonlinear effects are simplified as linear systems or neglected through processing. In this paper, a simplified model has been constructed for dual-beam CMFs based on the theory of nonlinear dynamics, with the spring–damper system as the medium for the dual-beam coupled vibrations. Next, the dynamics differential equation of the coupled vibrations is set up on the basis of the Lagrangian equation. Furthermore, numerical solutions are obtained using the Runge–Kutta fourth-order method. The study then fits discrete points of the numerical solutions, which are converted into the frequency domain to observe the existence of frequency-doubling signal components. Our findings show that frequency-doubling components exist in the spectrogram, proving that these nonlinear influences are a result of the motions of coupled vibrations. In this study, non-linear frequency-doubling signal sources are qualitatively analyzed to formulate a theoretical basis for CMFs design.

## 1. Introduction

Coriolis mass flowmeters (CMFs), which can directly measure the mass flow of fluids in high-precision instruments based on the Coriolis effect, have been applied extensively in the aviation, aerospace, petrochemical and food-processing industries [1,2]. Figure 1 shows the structure of a CMF with a U-type tube; the conventional method of calculating flow rate is to measure the time difference, i.e., phase difference between the waves detecting at point B and B’ [3,4], that can effectively reduce the sensitivity of external vibration, so it has been widely studied by scholars.

CMFs with measuring tubes that have small amplitudes are considered to be linear systems. However, it has been found paper that nonlinear components can be detected in the actual output signals. In 1998, Cheesewright et al. [5] found that there were components of excitation-doubling signals (twice and three times) in the frequency spectrum of the measuring tube vibration signals. In 2009, a study by Dezhi et al. [1] verified this phenomenon through theoretical analyses and experiments, as shown in Figure 2, and they further indicated that the impact of nonlinearity on the vibrations of CMF measuring tubes primarily manifested as additional vibration components multiplied by the excitation frequency. The additional vibration components would significantly distort the relationship between the output signal of CMF calibrated on frequency and the measured mass flow, thereby producing a relatively large measurement error.

Multiple studies have been conducted on the sources of these nonlinear phenomena. Cheesewright et al. [6] discussed how pulsating flow affected the results of measuring tubes. Belhadj et al. [7] simulated the pulsating flow of straight-tube and curved-tube CMFs with different geometric shapes using the ANSYS software. Kutin and Bajsić [8] examined the influence of non-ideal factors, including boundary distance, axial force, additional mass, damper and excitation, on CMFs with long straight tubes. Wang and Baker [9] simulated the influence of critical parameter changes on CMF measuring tubes due to manufacturing factors. Bobovnik et al. [10,11] first proposed to take into account a finite volume (FV)/finite element (FE) fully coupled numerical model of straight-tube CMFs, and studied the installation effects of CMFs with single long straight tubes and double U-shaped tubes using a fully coupled numerical model. Enz [12] studied the influence of position deviations of the actuators and detectors of CMF measuring tubes on the vibration phase shift of the tubes using multi-scale disturbance analysis. Svete et al. [13] pointed out that, in building the CMF numerical model, the continuity and momentum equation of pulsating flow should be accounted for, along with the limitations of the measuring tubes. Furthermore, Thomsen et al. [14] conducted a study on the bending vibrations of CMF measuring tubes, with a particular focus on the fluid flow and minor non-uniformity or asymmetry in stiffness, mass and damping, along with vibration-phase space displacements resulting from weak stiffness and damping nonlinearities. Bucolo et al. [15,16,17] researched the nonlinear analysis in microfluidics experimentation, and created a relationship between the nonlinear bubble flow (macroscopic dynamics) and the complex velocity patterns inside the bubbles (microscopic dynamics). In addition, most studies have focused on non-ideal boundary conditions [18,19,20,21], fluid features, fluid–structure interactions [22,23,24] and defects related to structural inhomogeneity [25]. In the past ten years, to the best of our knowledge, only a few studies have investigated the sources of nonlinearity. However, studies have shown that non-linear interference can be detected even in ‘’perfect’’ CMFs in the case of ‘’zero flow’’, for which no reasonable explanation has been offered to date.

Physically, a nonlinear phenomenon refers to a transition and sudden change from regular motion to irregular motion. A variety of nonlinear factors, such as physics, geometry, structure, dissipation, motion, and coupled factors, exist in all kinds of engineering systems. Therefore, it is inevitable that a system experiences complex motions beyond the linear range under certain conditions [26]. Typically, system motion resonance in a frequency-trapping phenomenon, or a ‘’synchronization phenomenon’’ [27,28], can be detected in a coupled nonlinear vibration system. No theoretical explanation has been offered for the mechanism of this nonlinear phenomenon in the sensitive units of resonant sensors. Therefore, this paper aims to build a theoretical correlation of the nonlinear mechanism of antiphase synchronization and dual-tube CMFs by conducting mechanism analysis on dual-tube CMFs from the perspective of coupled vibrations. The nonlinear analysis of single-beam forced vibrations in the second section can help prove that there is no frequency-doubling component during the zero-flow operation of any measuring tube in separate analyses. As a result, in the third section, the composition of the fundamental frequency and the frequency-doubling harmonic wave is obtained from the solutions of differential equations of coupled vibrations based on the established dual-beam coupled vibration model. In section four, significant frequency-doubling components are found in the numerical results when the coupled vibration equations are solved using the Runge–Kutta method. Finally, a method for improving the SNR of CMFs is proposed in controlled experiments; this can form the theoretical basis for improving the performance of CMF sensors.

## 2. Nonlinearity Analysis of Single-Beam Forced Vibration Model

It is important to simplify the model since dual-tube CMFs have complex tube structures and different supporting and fixation conditions. To determine the nonlinearity, this study excludes the influence of processing technologies, and the asymmetric distribution of mass and fluid discontinuity. First, it is assumed that the tube structures are highly symmetric. Specifically, the symmetric measuring tubes perform antiphase and synchronous motions around a fixed axis at zero flow [29]; therefore, the CMFs can be regarded as two thin-walled cantilever beams placed parallelly on the base. Second, this paper examines the nonlinearity resulting from coupling resonators, and thus only builds a vibration model in the empty-tube state. The study neglects the effects of viscous frictional forces and the Coriolis force in opposing directions when fluid flows in the double tubes.

According to the Huygens pendulum model, the operating platform of CMFs is considered to be a spring–damper system, which ignores the influence of the air friction caused by tube vibrations and the placement of excitation coils. Therefore, complex CMFs can be simplified as abstract models, as shown in Figure 3. $z_{1},z_{2},z$ represent the displacements of beam vibrations under the respective coordinate systems and the system displacements; $m_{1},m_{2},m$ stand for mass of the beams and the base, and $F_{1},F_{2}$ are the excitation forces of equal magnitude but in opposite directions.

The differential elements of the beams are analyzed to obtain the forces, as shown in Figure 4. 

Further, ($z_{1},z_{2},z$) are taken as the generalized coordinates, Q as the shear force and M as the bending moment, according to the D’Alembert’s principle. Considering the boundary conditions of cantilever beams, a D’Alambert equation with the mass of ρAdx can be constructed as follows:(1)$$\left(Q+\frac{\partial Q}{\partial x}dx\right)-Q+\frac{\partial^{2}z_{1}}{\partial t^{2}}\rho Adx-\frac{\partial^{2}z}{\partial t^{2}}\rho Adx=0$$

According to statics, the shear force and bending moment of the beam have the following relations:$$Q=\frac{\partial M}{\partial x},\;M=EJ\frac{\partial^{2}z_{1}}{\partial x^{2}}$$

The constraint conditions of a beam with one end firmly supported and one end free are as follows:$$\{\begin{array}{c} M\left(l,t\right)=0\\ Q\left(l,t\right)=-F\\ z_{1}\left(0,t\right)=0\\ \frac{\partial z_{1}\left(x,t\right)}{\partial x}{|}_{x=0}=0\\ \frac{\partial^{2}z_{1}\left(x,t\right)}{\partial x^{2}}{|}_{x=0}=0 \end{array}$$and obtain
(2)$$EJ\frac{\partial^{4}z_{i}}{\partial x^{4}}+\rho A\frac{\partial^{2}z_{i}}{\partial t^{2}}=\rho A\frac{\partial^{2}z}{\partial t^{2}}\;\;\;\;\;\;\;\;\;\;\left(i=1,2\right),$$where $\rho A\frac{\partial^{2}z}{\partial t^{2}}$ is the non-exciting external force from the base. The external excitation makes the beams vibrate during actual operation. The excitation force $F_{i}\left(t\right)$ acts on the endpoint of the tubes, and the distribution expressed by the function δ can be written as follows:(3)$$F_{i}\left(t\right)=F_{0}\left(t\right)\delta \left(x-l\right),$$

On this basis, the vibration differential equation of measuring tubes under external excitation is
(4)$$EJ\frac{\partial^{4}z_{i}}{\partial x^{4}}+\rho A\frac{\partial^{2}z_{i}}{\partial t^{2}}=F_{0}\left(t\right)\delta \left(x-l\right)+\rho A\frac{\partial^{2}z}{\partial t^{2}}\;\;\;\;\;\;\;\;\;\left(i=1,2\right),$$

To obtain the approximate solution of the motion equation, the paper discretizes the differential equation using the mode superposition method, as follows:(5)$$z_{i}\left(x,t\right)=\overset{\infty }{\underset{j=1}{\sum }}\emptyset_{j}\left(x\right)q_{j}\left(t\right)\;\;\;\;\;\;\;\;\;\;\;\left(i=1,2\right),$$where the value of $\emptyset_{j}\left(x\right)$, the j-order vibration mode function, is related to point positions on the beams and $q_{j}\left(t\right)$ are generalized coordinates only correlated with time. Based on the orthogonality condition of the modal analysis, the equation of j-order canonical coordinates can be derived as
(6)$${\ddot{q}}_{j}+w_{j}{}^{2}q_{j}=Q_{j}\left(t\right),$$where $Q_{j}\left(t\right)$ is the generalized force and can be expressed as
(7)$$Q_{j}\left(t\right)=\int_{0}^{l}\sin w_{F}t\cdot \delta \left(x-l\right)\emptyset_{j}\left(x\right)dx+\rho A\frac{\partial^{2}z}{\partial t^{2}}\emptyset_{j}\left(x\right)dx,$$

When (7) is substituted to (6), the expression for $q_{j}\left(t\right)$ is
(8)$$\begin{array}{ll} q_{j}\left(t\right)=\frac{1}{w_{j}}\int_{0}^{t}[ & Q_{j}\left(\tau \right)\sin[w_{j}\left(t-\tau \right)]dt=\frac{a\emptyset_{j}\left(l\right)}{w_{j}{}^{2}-w^{2}}\left(\sin wt-\frac{w}{w_{j}}\sin w_{j}t\right)\\ & +\rho A\dot{z}\left(1-\frac{\sin w_{j}t}{w_{j}}-\cos w_{j}t\right)\int_{0}^{l}\emptyset_{j}\left(x\right)dx \end{array}$$

According to Equations (5)–(8), the expression for $z_{i}\left(x,t\right)$ is
(9)$$\begin{array}{l} z_{i}\left(x,t\right)=\overset{\infty }{\underset{j=1}{\sum }}\frac{\emptyset_{j}\left(x\right)}{\rho Aw_{j}}[\emptyset_{j}\left(l\right)\frac{a}{w_{j}{}^{2}-w_{F}{}^{2}}(w_{j}\sin w_{F}t-w_{F}\sin w_{j}t)\\ +\rho A\dot{z}\left(1-\frac{\sin w_{j}t}{w_{j}}-\cos w_{j}t\right)\int_{0}^{l}\emptyset_{j}\left(x\right)dx]\;\left(i=1,2\right), \end{array}$$

If the excitation force is considered as the sole external force, and the influence of the base is neglected, the forced vibration modal response is
(10)$$q_{j}\left(t\right)=\frac{a\emptyset_{j}\left(l\right)}{w_{j}{}^{2}-w_{F}{}^{2}}\left(\sin w_{F}t-\frac{w_{F}}{w_{j}}\sin w_{j}t\right),$$


When $w_{F}=w_{1}$, the expression of the first-order vibration modal response is
(11)$$q_{1}\left(t\right)=\underset{w_{F}\to w_{1}}{\lim}\frac{a\emptyset_{1}\left(l\right)}{w_{j}{}^{2}-w_{F}{}^{2}}\left(\sin w_{F}t-\frac{w_{F}}{w_{j}}\sin w_{j}t\right)=a\emptyset_{1}\left(l\right)\left(\frac{\sin w_{1}t}{w_{1}{}^{2}}-\cos w_{1}t\right)/2$$

Further, the single-beam j-order vibration mode is typically expressed as
(12)$$\emptyset_{j}\left(x\right)=\cos\beta_{j}x-\cos h\beta_{j}x+\xi_{j}\sin\beta_{j}x-\sin h\beta_{j}x,$$where,
$$\xi_{j}=\frac{\cos\beta_{j}x+\cos h\beta_{j}x}{\sin\beta_{j}x+\sin h\beta_{j}x}\;,$$

Equation (11) is multiplied by Equation (12) to obtain the forced vibration response of the single-beam first-order vibration mode
(13)$$y_{1}\left(x,t\right)=a\emptyset_{1}\left(x\right)\emptyset_{1}\left(l\right)\left(\frac{\sin w_{1}t}{w_{1}{}^{2}}-\cos w_{1}t\right)/2,$$

The forced vibration responses of the second-order and multi-order vibration modes are
(14)$$y_{j}\left(x,t\right)=a\emptyset_{j}\left(x\right)\frac{\emptyset_{j}\left(l\right)}{w_{j}{}^{2}-w_{1}{}^{2}}\left(\sin w_{1}t-\frac{w}{w_{j}}\sin w_{j}t\right),$$


Based on the above derivations, the required signals or the components containing the first-order frequency can be obtained. Then, the response component of beam vibrations containing the first-order frequency is
(15)$$a_{1}\left(x,t\right)=a\left[\emptyset_{1}\left(x\right)\emptyset_{1}\left(l\right)\frac{\left(\frac{\sin w_{1}t}{w_{1}{}^{2}}-\cos w_{1}t\right)}{2}+\overset{\infty }{\underset{j=2}{\sum }}\frac{\emptyset_{j}\left(x\right)\emptyset_{j}\left(l\right)\sin w_{1}t}{w_{j}{}^{2}-w_{1}^{2}}\right],$$

Further, the response components including second-order and other multi-order frequencies can be expressed as
(16)$$a_{j}\left(x,t\right)=a\emptyset_{j}\left(x\right)\frac{\emptyset_{j}\left(l\right)}{\frac{w_{1}{}^{3}}{w_{j}}-w_{1}w_{j}}\sin w_{j}t,$$

This shows that when the beams experience first-order forced vibrations in the zero-flow state, no harmonic wave is found in the output frequency. In contrast, when there is flow, a non-doubled harmonic phenomenon will exist. This conclusion is similar to that drawn by Cheesewright et al. [6], in that the detector signal found during pulsating flow involves four components with different frequencies. However, it is apparent that this is not the source of the frequency-doubling signals in the zero-flow state. Hence, it is imperative to reanalyze the dual-tube CMF model.

## 3. Analysis of Coupled Vibrations of the Dual-Tube Model

In the previous section, a spring–damper model is established for dual-beam coupled vibrations. On this basis, the generalized coordinates of the system are redefined as $\left(q_{11},\;q_{21},\;z\right)$, and the vibration expression of tube i is represented by the following function:(17)$$z_{i}\approx \overset{3}{\underset{j=1}{\sum }}\emptyset_{j}\left(x\right)q_{ji}\left(t\right)\;\left(i=1,2\right),$$

Investigating the second-order and third-order vibration modals of continuously vibrating beams with infinite degrees of freedom is of great importance in the study of specific nonlinear influences [30], although the amount of the second-order and third-order frequency component amplitudes is smaller than that of the first-order component. Next, differential equations for system motion should be obtained via the Lagrangian equation [31], the general form of which can be written as follows:(18)$$\frac{d}{dt}\left\{\frac{\partial \left(T-U-U_{k}\right)}{\partial {\dot{q}}_{j}}\right\}-\frac{\partial \left(T-U-U_{k}\right)}{\partial q_{j}}=Q_{r},$$

The generalized coordinates are expressed as $q=\left(q_{1i},q_{2i},q_{3i}\ldots \ldots .,q_{ji},z\right)$, where i=1 and 2 denote the two beams and j represents the j-order modal. In addition, the change in the gravitational potential energy of the beams during deformation, along with the potential energy generated by the deformation of the Euler beams along the central axis, can be ignored according to the small-deformation theory. Therefore, the deformation energy of the beams during nonuniform bending is
(19)$$U=\frac{1}{2EJ}\int_{0}^{l}{\left[M\left(x\right)\right]}^{2}dx=\frac{EJ}{2}\int_{0}^{l}{\left(\frac{\partial^{2}z_{1}}{\partial x^{2}}\right)}^{2}dx+\frac{EJ}{2}\int_{0}^{l}{\left(\frac{\partial^{2}z_{2}}{\partial x^{2}}\right)}^{2}dx,$$

As can be seen from Figure 2, the bases of CMFs are simplified as spring–damper systems with a single degree of freedom. Therefore, the potential energy of the base is the elastic potential energy of the spring
(20)$$U_{k}=\frac{1}{2}kz^{2},$$

The kinetic energy of all the system parts is added to obtain the total kinetic energy, as follows:
(21)$$T=\frac{1}{2}m{\left(\frac{\partial z}{\partial t}\right)}^{2}+\frac{1}{2}\frac{m_{1}}{l}\int_{0}^{l}{\left(\frac{\partial z}{\partial t}-\frac{\partial z_{1}}{\partial t}\right)}^{2}dx+\frac{1}{2}\frac{m_{2}}{l}\int_{0}^{l}{\left(\frac{\partial z}{\partial t}-\frac{\partial z_{2}}{\partial t}\right)}^{2}dx,$$


The generalized force $Q_{r}$ of the system comprises the following components:$$Q_{z}=-c\frac{\partial z}{\partial t}\;\mathrm{and}\;Q_{z_{i}}=F_{0}\left(t\right)\delta \left(x-l\right)$$

(1) For the generalized coordinates $q_{i1}$,
(22)$$\begin{array}{ll} \frac{d}{dt}\left\{\frac{\partial \left(T-U-U_{k}\right)}{\partial \dot{q_{i1}}}\right\} & =\frac{d}{dt}\left\{\frac{1}{2}\frac{m_{i}}{l}\int_{0}^{l}\left[2{\left(-1\right)}^{i}\emptyset_{1}\left(x\right)\frac{\partial z}{\partial t}+2\emptyset_{1}{}^{2}\left(x\right)\frac{\partial q_{i1}\left(t\right)}{\partial t}\right]dx\right\}\\ & =\frac{m_{i}}{l}\int_{0}^{l}\left[{\left(-1\right)}^{i}\emptyset_{1}\left(x\right)\frac{\partial^{2}z}{\partial t^{2}}+\emptyset_{1}{}^{2}\left(x\right)\frac{\partial^{2}q_{i1}\left(t\right)}{\partial t^{2}}\right]dx, \end{array}$$(23)$$\begin{array}{ll} \frac{\partial \left(T-U-U_{k}\right)}{\partial q_{i1}} & =-\frac{\partial \left(\;\sum_{i=1}^{2}\frac{EJ}{2}\int_{0}^{l}{\left(\frac{\partial^{2}\emptyset_{1}\left(x\right)q_{i1}\left(t\right)}{\partial x^{2}}\right)}^{2}dx\right)}{\partial q_{i1}}\\ & =-EJq_{i1}\left(t\right)\int_{0}^{l}{\left(\frac{\partial^{2}\emptyset_{1}\left(x\right)}{\partial x^{2}}\right)}^{2}dx, \end{array}$$

Equations (22) and (23) can be substituted into Equation (18) to express the differential equation of motion as follows:(24)$$\frac{m_{i}}{l}\int_{0}^{l}\left[{\left(-1\right)}^{i}\emptyset_{1}\left(x\right)\frac{\partial^{2}z}{\partial t^{2}}+\emptyset_{1}{}^{2}\left(x\right)\frac{\partial^{2}q_{1i}\left(t\right)}{\partial t^{2}}\right]dx+EJq_{1i}\left(t\right)\int_{0}^{l}{\left(\frac{\partial^{2}\emptyset_{1}\left(x\right)}{\partial x^{2}}\right)}^{2}dx=Q_{z_{i}},$$

(2) For the generalized coordinates z,
(25)$$\frac{d}{dt}\left\{\frac{\partial \left(T-U-U_{k}\right)}{\partial \dot{z}}\right\}=\left(m+m_{1}+m_{2}\right)\frac{\partial^{2}z}{\partial t^{2}}+\overset{2}{\underset{i=1}{\sum }}\frac{m_{i}}{l}\int_{0}^{l}{\left(-1\right)}^{i}\emptyset_{1}\left(x\right)\frac{\partial^{2}q_{i1}\left(t\right)}{\partial t^{2}}dx,$$(26)$$\frac{\partial \left(T-U-U_{k}\right)}{\partial q_{z}}=-kz,$$

Equations (25) and (26) are substituted into Equation (18) to obtain the differential equation of motion, as follows:(27)$$\left(m+m_{1}+m_{2}\right)\frac{\partial^{2}z}{\partial t^{2}}+\overset{2}{\underset{i=1}{\sum }}\frac{m_{i}}{l}\int_{0}^{l}{\left(-1\right)}^{i}\emptyset_{1}\left(x\right)\frac{\partial^{2}q_{i1}\left(t\right)}{\partial t^{2}}dx+kz=Q_{z},$$

Since the physical parameters and vibration mode functions are known for the differential equation formed by combining Equations (24) and (27), the differential equation can be solved.

Finally, the system of dynamics differential equations of the dual-beam coupled vibration model can be obtained as follows:(28)$$\{\begin{array}{c} \frac{m_{i}}{l}\int_{0}^{l}\left[{\left(-1\right)}^{i}\emptyset_{1}\left(x\right)\frac{\partial^{2}z}{\partial t^{2}}+\emptyset_{1}{}^{2}\left(x\right)\frac{\partial^{2}q_{1i}\left(t\right)}{\partial t^{2}}\right]dx+EJq_{1i}\left(t\right)\int_{0}^{l}{\left(\frac{\partial^{2}\emptyset_{1}\left(x\right)}{\partial x^{2}}\right)}^{2}dx=Q_{z_{i}}\\ \left(m+m_{1}+m_{2}\right)\frac{\partial^{2}z}{\partial t^{2}}+\overset{2}{\underset{i=1}{\sum }}\frac{m_{i}}{l}\int_{0}^{l}{\left(-1\right)}^{i}\emptyset_{1}\left(x\right)\frac{\partial^{2}q_{i1}\left(t\right)}{\partial t^{2}}dx+kz=-c\frac{\partial z}{\partial t} \end{array},$$

The equations are simplified for simplicity, setting certain conditions as follows:$$\int_{0}^{l}\emptyset_{1}\left(x\right)dx=W_{1},$$$$\int_{0}^{l}\emptyset_{1}{}^{2}\left(x\right)dx=W_{2},$$$$\int_{0}^{l}{\left(\frac{\partial^{2}\emptyset_{1}\left(x\right)}{\partial x^{2}}\right)}^{2}dx=W_{3},$$

By substituting them into Equation (16), the final form of the dynamics equation is found to be
(29)$$\{\begin{array}{c} \rho A\left[{\left(-1\right)}^{i}W_{1}\ddot{z}+W_{2}\ddot{q_{1i}}\right]+EJq_{1i}W_{3}=Q_{z_{i}}\\ \left(m+m_{1}+m_{2}\right)\ddot{z}+W_{1}\overset{2}{\underset{i=1}{\sum }}\rho A{\left(-1\right)}^{i}\ddot{q_{1i}}+kz+c\dot{z}=0 \end{array},$$i=1,2

## 4. Numerical Solutions

As the dynamics differential equation system in Equation (17) has high-order differentials and is a non-linear ordinary differential equation, its general solutions cannot be solved via mathematical calculations. To acquire the response $q_{1i}$ of the first-order vibration modal, this study estimates the solutions approximately using numerical solutions. The Runge–Kutta methods are used for numerical solutions. First of all, the equations are reduced in order, so that the dynamics equations with high-order differentials are applicable to numerical solutions. First, set
$$z=y1,\;\dot{z}=y2,\;q_{11}=y3,\;\dot{q_{11}}=y4,\;q_{12}=y5,\;\dot{q_{12}}=y6$$

The expression of the differential equations can then be simplified as
(30)$$\{\begin{array}{c} y2=\dot{y1}\\ y4=\dot{y3}\\ y6=\dot{y5}\\ \left(M+m_{1}+m_{2}\right)\dot{y2}+W_{1}\rho A\left[\dot{y\left(6\right)}-\dot{y\left(4\right)}\right]+Ky1+cy2=0\\ \dot{y2}=\frac{\left[\frac{F_{i}-EJy\left(a\right)W_{3}}{\rho A}-W_{2}\dot{y\left(b\right)}\right]}{{\left(-1\right)}^{i}W_{1}} \end{array},$$where
$$\{\begin{array}{c} i=1,\;a=3,\;b=4\\ i=2,\;a=5,\;b=6 \end{array},$$


The six reduced-order differential equations in Equation (18) are solved by the Runge–Kutta methods using MATLAB. According to actual CMF parameters, the parameters of the coupled tubes are listed in Table 1:

The numerical solutions for $q_{11}$ and $q_{21}$ are obtained. If $q=q_{11}-q_{21}$, a series of discrete points of the numerical solutions for q can be acquired. Furthermore, the discrete points are fitted using the method of cubic spline interpolation for solving a continuous curve. Moreover, the curve is sampled at a sampling frequency Fs = 2400 Hz, and the number of points L = 48,001. Upon applying the fast Fourier transform (FFT), the spectrogram of the response of the dual-beam coupled vibration system can be found, as shown in Figure 5.

According to Figure 5, the main frequency component $\Omega_{1}$ is 124.2 Hz, which is consistent with the CMF’s first-order inherent frequency. Further, other signals can be observed in the spectrogram. Specifically, components exist when $\Omega_{2}$ = 372.4 Hz, $\Omega_{3}$ = 620.9 Hz, $\Omega_{4}$ = 868.9 Hz and $\Omega_{5}$ = 1118 Hz. In addition, the frequencies of these components correspond to 3, 5, 7 and 9 times of the main frequency component, respectively. Thus, the existence of the frequency-doubling signals can be verified through theoretical derivations. This finding indicates that the coupling between the two measuring tubes due to the vibration of the entire system is correlated with the influence of nonlinearities. Further, the frequency doubling can be detected in the final signal because of the superposition of the coupling effect.

It is important to note that the amplitudes of multiplication components decrease with the signal frequency in the CMFs [1,5], as shown by $\omega_{0}$, 2$\omega_{0}$ and 3$\omega_{0}$ in Figure 2 and as mentioned in a previous report. Further, only odd multiples of frequencies, such as 3 times and 5 times, can be found in the spectrogram solved by the dynamics equation in the simplified model, while even multiples are difficult to detect. One reason for this could be that the proposed simplified model of straight tube coupled vibrations differs from the CMF model with two U-shaped tubes, which has a spatially-symmetrical straight tube and connecting tube, leading to a more complex vibration mode function. The coupled vibrations in such a symmetrical structure, with the superimposition of the coupled vibrations of the connecting tube, could exacerbate the noise generated by nonlinearities. Therefore, the next step is to improve the model.

## 5. Nonlinearity Suppression Analysis

In previous sections, the numerical solution results validated the reliability of the coupled vibration model. In this section, the mass of the base and the magnitude of the excitation are studied to offer new perspectives for suppressing nonlinearities, enhancing the design of the CMF measuring tube system.

(a) Varying the excitation force magnitude

Spectrograms of various excitation forces (0.5 N, 1.5 N and 3 N) are shown in Figure 6, from which it can be seen that the frequency-doubling components increase with increasing excitation force magnitudes. Moreover, the greater the excitation force, the less obvious the increasing trend. Although the amplitude of the frequency-doubling signal increases, the same cannot be said for the SNR. It can be seen in Figure 6c that the peak signal value for 3 N excitation is close to 30 dB, while the peak of the frequency-doubling component is close to −50 dB. The presented SNR is better than that of the 0.5 N excitation force shown in the figure, indicating that greater excitation forces can increase the components created by nonlinearities; however, this proportional trend decreases in comparison to the desired gain signal. Therefore, selecting an appropriate excitation force can effectively reduce the influence of nonlinearities. However, this is not an optimal solution, since the improvement is determined by beams with different physical parameters. In other words, this method cannot ensure a complete and effective reduction of the influence of nonlinearities.

(b) Changing the mass of the measuring tubes

It can be seen from Figure 7 that the resonant frequencies decrease with increasing mass. Accordingly, the frequency decreases from 148.1 Hz to 104.7 Hz, and finally to 85.47 Hz. With the increasing mass of the beams, the curve gradually becomes smoother, while the frequency-doubling signal components gradually decrease. This trend indicates that the changes (only the beam thickness is changed in this section n, while the length and cross-section of the beams remain constant) in the mass of the beams can suppress the nonlinearities to some extent. However, the processing cost of the measuring tubes is generally high, a cost that will be increased by attempting to improve the processing technique. Further, improvements to the measuring tubes fail to fundamentally suppress the nonlinearities, as the frequency-doubling signals still exist and their influence on the SNR is non-negligible.

## 6. Conclusions

In this study, based on the synchronization theory, the complex CMF model is simplified to a beam model for easier calculations. Focusing on a single beam, this study applies the modal analysis method to derive the kinematic equation for a single beam under the condition of sinusoidal excitation. The equation verifies that no frequency-doubling signal is detected in the output frequency when the beam is under first-order forced vibration under no-flow conditions and only high-order modal components exist, influencing the SNR. The verification indicates that a non-frequency-doubling harmonic component may exist in the zero-flow state. Based on a theoretical analysis of dual-beam coupled vibrations, the dynamics differential equations of the coupled vibrations are built using the Lagrangian equation. Next, the solutions of the differential equations are analyzed using the Runge–Kutta methods, and discrete points of the numerical solutions are fitted and transformed into the frequency domain through FFT, to facilitate observation. A significant odd frequency-doubling phenomenon can be observed in the spectrogram, proving that the frequency-doubling phenomenon does exist in double beams under the influence of coupled vibrations.

To suppress the nonlinearities, this study conducts a controlled qualitative analysis of the influences of several physical parameters of the system on the frequency-doubling components generated by the coupled vibrations. Therefore, recommendations can be offered to improve the design of CMFs. Furthermore, this paper also provides a theoretical basis and reference for discussing how to fundamentally suppress nonlinearities introduced by coupled vibrations.

## Figures and Tables

**Figure 1 sensors-20-06340-f001:**
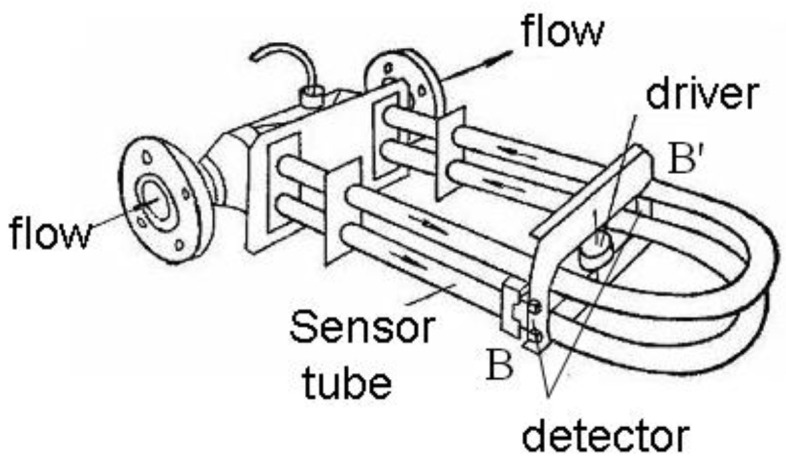
Structure of Coriolis mass flowmeter (CMF) with U-type tube.

**Figure 2 sensors-20-06340-f002:**
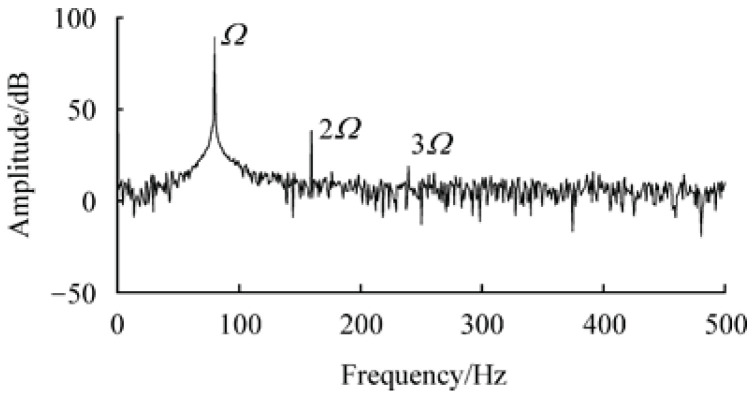
Vibration frequency spectrum of CMF measuring tubes during closed-loop operation [1].

**Figure 3 sensors-20-06340-f003:**
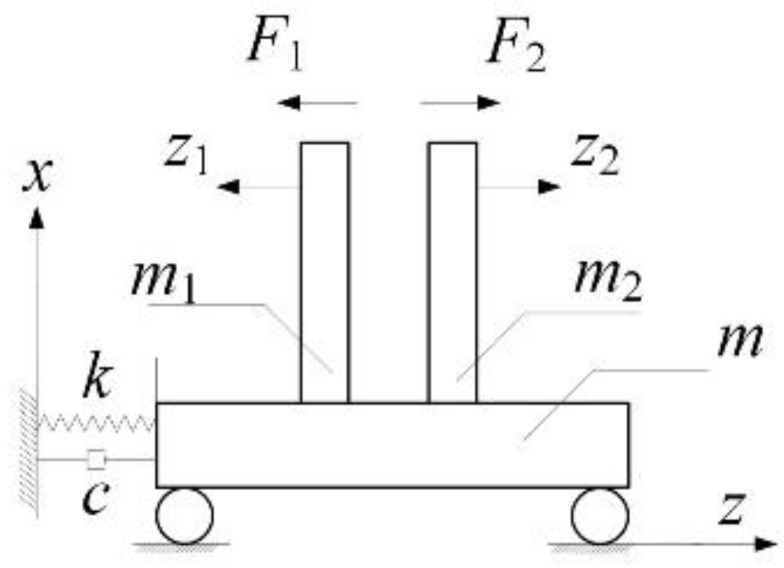
Dual-beam coupled vibration model based on simplified CMFs.

**Figure 4 sensors-20-06340-f004:**
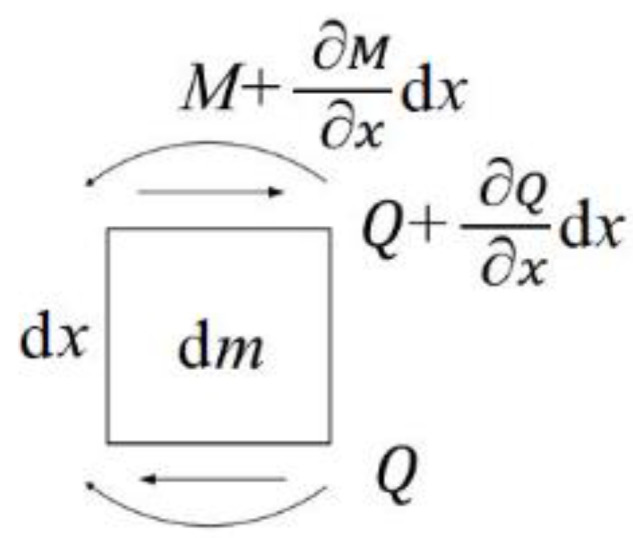
Force of beam differential elements.

**Figure 5 sensors-20-06340-f005:**
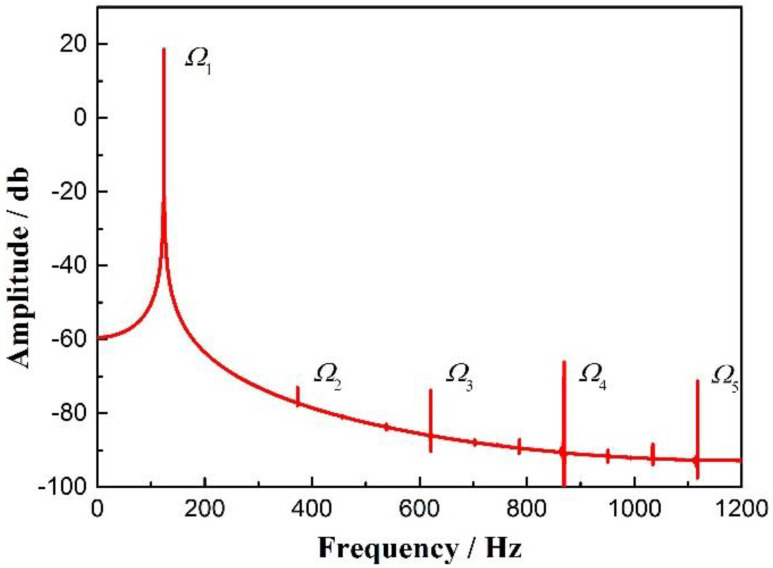
Spectrogram of the response of the dual-beam coupled vibration model.

**Figure 6 sensors-20-06340-f006:**
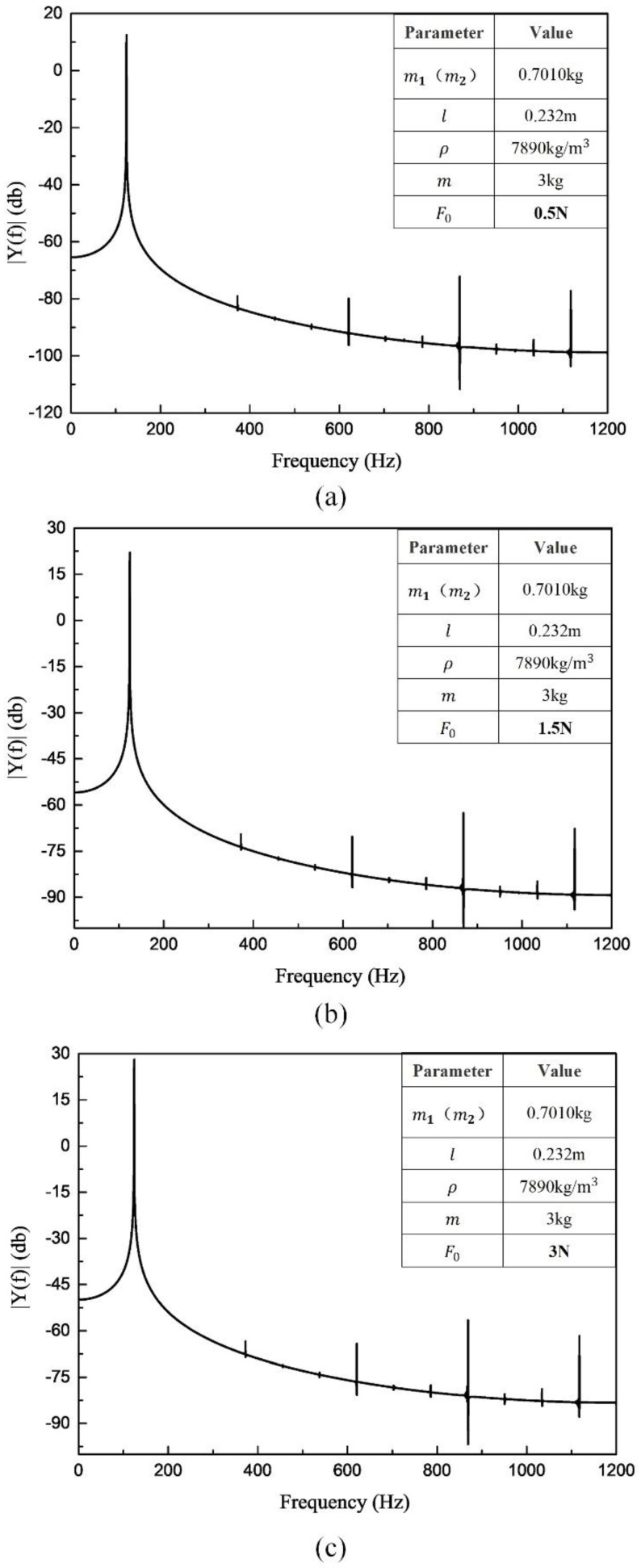
Spectrogram when changing the excitation force ((**a**)$\;F_{0}$ = 0.5 N, (**b**)$\;F_{0}$ = 1.5 N, (**c**)$\;F_{0}$ = 3 N) for a single beam.

**Figure 7 sensors-20-06340-f007:**
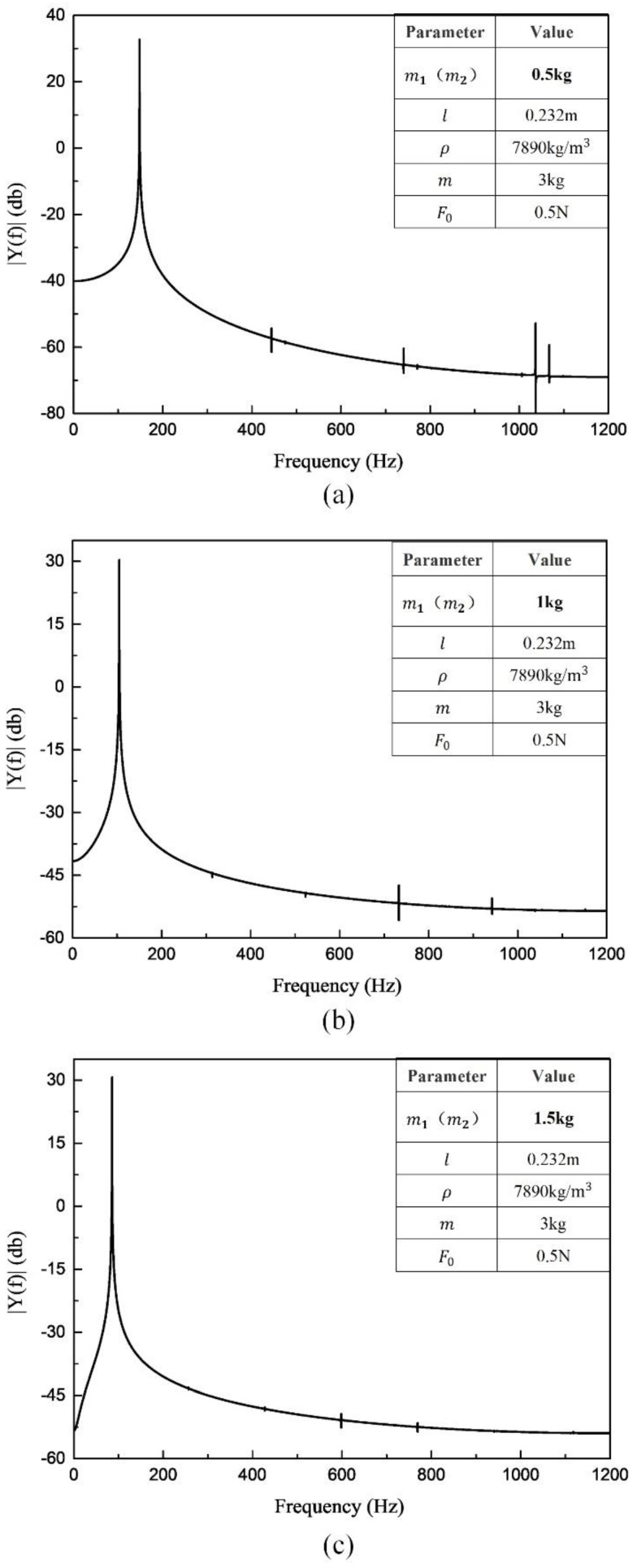
Spectrogram when changing the mass ((**a**)$\;m_{i}$ = 0.5 kg, (**b**) $m_{i}$ = 1 kg, (**c**) $m_{i}$ = 1.5 kg).

**Table 1 sensors-20-06340-t001:** Parameters of measuring tubes.

Parameter	Value
Mass of a single beam	0.7010 kg
Length of a single beam	0.232 m
Diameter of a single beam	0.5 cm
Density	7890 kg/$m^{3}$
Mass of the base	3 kg
Elastic modulus	200 GPa

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
