# Peer review of "Synchronization Theory-Based Analysis of Coupled Vibrations of Dual-Tube Coriolis Mass Flowmeters"

_sensors, 2020, doi:10.3390/s20216340_

Round 1
Reviewer 1 Report
The paper is very well-written with good English. Presented is a mathematical model of the non-linear effects that cause higher harmonics in Coriolis mass flow sensors (CMFS). The paper starts with an excellent introduction with a thorough review of previous work on this topic. Although the paper is of good quality, a minor revision based on the items below would improve it significantly.
- The paper is quiet difficult to read and understand. The writing, structure and typesetting of the equations are perfect, but a bit more explanation in the mathematical derivation (and relation to the physics) would help the reader in understanding.
- Maybe add a figure (illustration) of a dual-tube CMFS as this might not be trivial for all readers. This also defines what the tube shape is (straight, triangular, U-shaped, ...), as that also varied a lot in many publications on this topic last years.
- There are no experiments or other way of verification of the theory. The model has such complexity that it fills a complete letter. However, it would be a good (or almost necessary) addition when experimental results proof the theory in a (future) publication. A comparison with the experimental results of a previously published paper might also be a good addition.
- In the section 'Nonlinearity Suppression Analysis', SNR (signal to noise ratio) is probably not the right term to use, because the peaks are not actual noise. Probably the term SINAD (signal to noise and distortion ratio) or another term covers this phenomenon better as distortion and non-linearity are more related than noise and non-linearity. In the rest of the paper, the term noise should also be replaced by nonlinearity.
- There are two sections with number 3: between sections 4 and 6, there is a second section 3.
- The paper misses a (concluding) answer on the following question: how do the nonlinearities influence the performance of the Coriolis mass flow sensor in the end? I.e., is the e.g. accuracy, sensitivity, linearity or range limited by the higher order harmonics? By lock-in amplification or filtering, the influence of the higher order harmonics can maybe simply be reduced?
- Maybe leave out 'Sensitive Units' from the title: it only makes the title longer without making it more clear.
Author Response
Point 1: The paper is quiet difficult to read and understand. The writing, structure and typesetting of the equations are perfect, but a bit more explanation in the mathematical derivation (and relation to the physics) would help the reader in understanding.

Response 1: For a better understanding, I add some derivation to the manuscript in section 2 (in red, line 115-120 &138-140). The derivation in section 3 is complete.
Point 2: Maybe add a figure (illustration) of a dual-tube CMFS as this might not be trivial for all readers. This also defines what the tube shape is (straight, triangular, U-shaped, ...), as that also varied a lot in many publications on this topic last years.
Response 2: I add an introduction and illustration of CMF in section 1 (in red, line 30-35)
Point 3: There are no experiments or other way of verification of the theory. The model has such complexity that it fills a complete letter. However, it would be a good (or almost necessary) addition when experimental results proof the theory in a (future) publication. A comparison with the experimental results of a previously published paper might also be a good addition.
Response 3: This paper focuses on demonstrating the frequency-doubling signals caused by coupled vibration, and carries out numerical calculation. The calculated results are basically consistent with the phenomenon obtained by previous researchers [as shown in reference 1,3]. At present, relevant experiments are being carried out.
Point 4: There are two sections with number 3: between sections 4 and 6, there is a second section 3.
Response 4: This is my fault, sorry for the trouble caused to your reading, I have corrected the correct number (in red, 5. Nonlinearity Suppression Analysis).
Point 5: The paper misses a (concluding) answer on the following question: how do the nonlinearities influence the performance of the Coriolis mass flow sensor in the end? I.e., is the e.g. accuracy, sensitivity, linearity or range limited by the higher order harmonics? By lock-in amplification or filtering, the influence of the higher order harmonics can maybe simply be reduced?
Response 5: Existing studies have elaborated on the effects of this nonlinearity, and I have added its effects in section1(in red, line 43-45), but it's not specific enough. The mathematical model I established in this paper is used to guide how to suppress this nonlinear phenomenon. In the next step, I will carry out experiments to obtain detailed inhibitory effects.
Point 6: Maybe leave out 'Sensitive Units' from the title: it only makes the title longer without making it more clear.
Response 6: I have modified the title according to your suggestion
Reviewer 2 Report
The paper presents new results about the flow meter based on Coriolis principles. The sensor device is very useful and widely used. The problem posed by the authors is very important: it consists in eliminating the nonlinearity of the system that lead to not precise measurements in the flowmeters. The approach in the paper is based on the principles of non linear dynamics and are related to the synchronization concepts.
This is original. Moreover the analogy with a dual beam coupled system is impressive and allows to make the presented approach very suitable to be investigated with appropriate mathematical model and simulation algorithms.
The mathematical approach is sufficiently well studied. The numerical discussion of Section 3 is fine and correct,
There is a mistake in indicating Section 3…..again…..it should be Section 5 (3. Nonlinearity Suppression Analysis). In this section the suppression of the nonlinearity is studied).
The numerical results indicate the suitability of the approach and are hopeful for the real application in the realization of the compensated results.
The importance of the study is today also for the microfluidic devices where compact flow meters must be integrated and where the precision is crucial, I therefore suggest to include in the reference the following contribution:
Microfluidic circuits and systems, IEEE circuits and systems magazine, 2009, 9(3), 6-19
Real-time detection of slug velocity in microchannels, Micomachines, 2020, 11(3), 241
Micro-optofluidic switch realized by 3D printing technology, Microfluidics and Nanofludidics, 2016, 20(4), 61
I encourage this research.
Author Response
Point 1: There is a mistake in indicating Section 3…..again…..it should be Section 5 (3. Nonlinearity Suppression Analysis). In this section the suppression of the nonlinearity is studied).
Response 1: This is my fault, sorry for the trouble caused to your reading, I have corrected the correct number (in red, 5. Nonlinearity Suppression Analysis).
Point 2: The importance of the study is today also for the microfluidic devices where compact flow meters must be integrated and where the precision is crucial.
Response 2: Thank you for your valuable advice. In recent years, many researches on coupling phenomenon have appeared in the field of MEMS. It is necessary to enrich the research background in Section 1 (in red, line 64-67)
